# Deep Learning-Based Advances in Protein Structure Prediction

**DOI:** 10.3390/ijms22115553

**Published:** 2021-05-24

**Authors:** Subash C. Pakhrin, Bikash Shrestha, Badri Adhikari, Dukka B. KC

**Affiliations:** 1Department of Electrical Engineering and Computer Science, Wichita State University, Wichita, KS 67260, USA; scpakhrin@shockers.wichita.edu; 2Department of Computer Science, University of Missouri-St. Louis, St. Louis, MO 63121, USA; bsmmy@mail.umsl.edu

**Keywords:** protein structure prediction, deep learning, protein contact map prediction, protein distance prediction, protein quality assessment

## Abstract

Obtaining an accurate description of protein structure is a fundamental step toward understanding the underpinning of biology. Although recent advances in experimental approaches have greatly enhanced our capabilities to experimentally determine protein structures, the gap between the number of protein sequences and known protein structures is ever increasing. Computational protein structure prediction is one of the ways to fill this gap. Recently, the protein structure prediction field has witnessed a lot of advances due to Deep Learning (DL)-based approaches as evidenced by the success of AlphaFold2 in the most recent Critical Assessment of protein Structure Prediction (CASP14). In this article, we highlight important milestones and progresses in the field of protein structure prediction due to DL-based methods as observed in CASP experiments. We describe advances in various steps of protein structure prediction pipeline viz. protein contact map prediction, protein distogram prediction, protein real-valued distance prediction, and Quality Assessment/refinement. We also highlight some end-to-end DL-based approaches for protein structure prediction approaches. Additionally, as there have been some recent DL-based advances in protein structure determination using Cryo-Electron (Cryo-EM) microscopy based, we also highlight some of the important progress in the field. Finally, we provide an outlook and possible future research directions for DL-based approaches in the protein structure prediction arena.

## 1. Introduction

Obtaining accurate descriptions of protein structures is a fundamental step toward understanding the underpinning of biology. Though there has been continuous improvement in the experimental approaches (X-ray Crystallography, Nuclear Magnetic Resonance (NMR) spectroscopy, Cryogenic Electron Microscopy (Cryo-EM), and others) for determining protein structures, the gap between the number of protein sequences and known structures is ever increasing. As of March 2021, the total number of protein structures deposited in Protein Data Bank (PDB) [1] is around 180 thousands, whereas the number of protein sequences deposited at the end of 2020 in Uniport/TrEMBL [2] is ≈207 millions. In that regard, protein structure prediction is one of the important problems in computational structural biology.

On the other hand, Deep Learning (DL) is becoming one of the mainstream technologies for various scientific application domains, including computer vision [3], natural language processing [4,5], speech recognition [6], and autonomous driving [5,7], among others [7]. In that regard, with the recent advancements in DL algorithms and the exponential growth in computational power, the field of protein structure prediction has also witnessed tremendous advances.

The Critical Assessment of protein Structure Prediction (CASP) assesses the state of the art in modeling protein structure from amino acid sequence. The first CASP competition, CASP1, was held in 1994. Fast forward, around the time of the CASP12 competition in 2016, contact prediction emerged as the key intermediate step toward accurate structure prediction. For the first time, a deep learning-based method, Raptor-X [8], achieved around 50% precision when evaluating top L/5 long-range predictions—almost twice as much precision compared to the CASP11 competition. Here L represents the length of the protein sequence. Soon after CASP12, a much improved version of the Raptor-X method [9] was released, and a fully open-source deep learning method DNCON2 [10], demonstrating a similar performance, was also released. After the results of the CASP13 competition were announced in 2018, the top performing methods including Raptor-X [11] and AlphaFold [12] had upgraded their methods to predict ‘distograms’ instead of just contacts. Different from an inter-residue contact, which only defines if a residue pair is less than or more than 8 Å, a distogram can define if the same pair falls within a smaller distance range. Hence, distogram predictions provide much richer information for 3-dimensional (3D) modeling. In the same competition, an end-to-end method [13] introduced by AlQuraishi [14] demonstrated a competitive performance and inspired the development of end-to-end deep learning methods. 

After the CASP13 competition, some groups continued the distogram prediction efforts. Inspired by the success of AlphaFold [12] but frustrated with the fact that AlphaFold was not fully open, Corte’s group developed and released an open-source implementation of AlphaFold’s method called ProSPr [15]. The trRosetta method, in particular, showed performance similar to AlphaFold and released an open-source method for distogram prediction. However, many others focused on real-valued distance prediction and developed methods such as DeepDist [16], RealDist [17], and the Generative Adversarial Network (GAN)-based method [18]. An open-source framework for distance prediction, PDNET [19], was also released around the same time.

After the CASP13 competition, it was also evident that evolutionary information captured in multiple sequence alignments (MSAs) was the most important input for structure prediction, and all research groups have their own in-house methods for generating MSAs. Unfortunately, the problem of generating high-quality alignments, particularly for difficult sequences, stands as a huge challenge. As an effort to build a single pipeline for generating high-quality and deep alignments and also to find remotely homologous sequences in the case of new (difficult) sequences, Zhang’s group developed and released DeepMSA [20].

There have been significant progresses in the field of protein structure prediction, especially those related to free modeling methods that generate structure models without homologous templates. Refer to some excellent reviews [21,22] for the details. Protein structure prediction can be classified into various categories: (a) One-Dimensional (1D) protein structure prediction, (b) Two-Dimensional (2D) protein structure prediction, and (c) 3D protein structure prediction.

1D structures of a protein are residue-wise quantities or symbols onto which some features of the native 3D structure are projected. Some examples of 1D structures are secondary structure assignment, solvent accessibility, etc. The 2D structures of a protein are contact maps and distance maps. There have also been recent review articles highlighting the DL methods in protein structure prediction [23,24]. Torrisi et al. [23] recently reviewed DL-based approaches for 1D protein structural annotations and methods for 2D protein structural annotations. Gao et al. [24] reviewed some advances in DL-based approaches for the protein sequence–structure–function paradigm. Although these are excellent reviews for some recent advances in some aspects of protein structure prediction, there is no comprehensive review in advances of DL-based approaches for protein structure prediction as observed in various CASP experiments. In addition, there is no comprehensive review that focuses on the DL-based advances in various steps of the protein structure prediction pipeline. This review aims to be useful to researchers who are interested in Deep Learning and its application to protein structure prediction.

In this article, we will highlight the recent developments in the application of Deep Learning for 3D protein structure prediction with a focus on various steps of the protein structure prediction pipeline. For template-free structure prediction approaches, the important steps are as follows: (i) identification of sequence homologs and generation of multiple sequence alignment, (ii) residue–residue contact prediction and/or residue–distance prediction, (iii) iterative fragment assembly simulations guided by potential or ab initio 3D modeling driven by contacts/distances, and (iv) atomic-level structure refinement and ranking or Quality Assessment of models [22,25] as shown in Figure 1. Although there is a little variation in steps in these prediction protocols, most successful protein structure prediction pipelines such as I-TASSER [26] are hybrid (combination of template-based structure prediction and template-free structure prediction); Rosetta [27] and RaptorX [28] can be thought of as having these important common steps.

In addition, recent advances in microscopy as well as algorithms for image processing have helped Cryo-EM become one of the most widely used techniques for the determination of protein structures and complexes. Figure 2 shows the cumulative number of electron maps released in Electron Microscopy Data Bank (EMDB) [29]. Although the number of these maps has significantly increased, there is an intermediate computational step that is required to obtain molecular structures from these Cryo-EM maps. There have been some remarkable DL-based advances for Cryo-EM-based protein structure prediction.

In this review, we highlight DL-based advances in each step of the protein structure prediction pipeline viz. advances in MSA generation, contact map prediction, protein residue–distance prediction, potentials to guide iterative fragment assembly, models, or quality assessment (QA), advances in overall protein prediction pipelines, and advances in Cryo-EM based protein structure determination and the future outlook for the protein structure prediction field.

## 2. Deep Learning-Based Advances in Various Steps of Protein Structure Prediction Pipeline

In this section, we highlight DL-based advances in various steps (Figure 1) of the protein structure prediction pipeline viz. multiple sequence alignment, contact map, distogram or real-value distance prediction, model quality assessment (QA) and refinement.

### 2.1. Advances in Approaches for Multiple Sequence Alignment

The generation of multiple sequence alignment (MSA) for a query protein is the first step in the majority of protein structure prediction pipelines [12,26,27,30]. Since the subsequent steps of contact prediction, distogram prediction, or real-valued distance prediction rely on the quality of the MSA, the generation of deep and high-quality MSA is of paramount importance for protein structure pipeline. Even in the light of AlphaFold2 (an end-to-end protein structure pipeline using Deep Learning that takes MSA as the input), the quality of MSA becomes increasingly important. Although not based on a Deep Learning-based approach, DeepMSA [20] represents one of the significant recent trends to improve the process of MSA creation. DeepMSA is a composite approach to generate MSA with large alignment depth and diverse sequence sources by merging sequences from whole-genome sequence databases and from metagenome databases. Since DeepMSA is the only method that allows the integration of multiple sequence searching tools (such as HHblits [30] and JackHMMER [31]) and multiple databases including the large metagenomic sequence databases (often helpful for difficult prediction cases), it was used by many CASP14 participants as well.

Next, we highlight DL-based advances in protein contact map prediction.

### 2.2. DL-Based Advances in Protein Contact Map Prediction

A pair of amino acids is said to be in contact if the distance between their carbon-beta atoms (carbon-alpha in case of glycine) is less than or equal to 8 Å (i.e., C_β_-C_β_ distance < 8 Å). Residue pairs in contact capture the possible physical interactions between the two residues. A true contact map for a protein with known structure is a 2D map of size “L × L” with 1 s at the locations where the distance is less than or equal to 8 Å and is 0s at all other cells. Here, L is the length of the corresponding protein sequence. Hence, for a protein sequence, predicting a protein contact map is predicting the contact probability for every possible inter-residue pair. From the perspective of reconstruction (building 3D models), not all contacts are equally useful. One technique to classify contacts is to group them into four categories based on how far apart the two residues are in the protein sequence: local (separated by less than six residues in the sequence), short-range (separated by six to 11), medium-range (separated by 12 to 23 residues), and long-range (24+ sequence separation). While local contacts capture secondary structure information, medium-range and long-range contacts are found to be useful for recovering the fold/shape accurately. Hence, the main focus in contact prediction is to predict medium-range and long-range contacts as accurately as possible. In addition, an entire contact map is not required for reconstructing a 3D structure [32]. Hence, the performance of a method that predicts contacts is evaluated based on the precision of top L or top 2 L most confident predictions. The predicted contacts have a confidence score associated with each pair, and this confidence score is used to rank the predicted contacts. From these top-ranked predicted contacts, usually, local and short-range contacts are ignored and the top L, or L/2, or L/5 contacts are evaluated by checking if these pairs are in contact or not in the true 3D structure.

From the perspective of machine learning, the contact prediction problem in protein structure prediction may be compared to the image segmentation problem in computer vision. In image segmentation, the input is an image of dimensions H × W × Z, where H, W, and Z are the height, width, and channels of the image respectively, and the output is a 2D matrix of size H × W, where each pixel either belongs to the object or not. Similarly, in the protein contact prediction problem, the input is the protein features of size L × L × N, where L is the length of the protein sequence and N is the number of channels, and the output is the probability values of the contacts as a 2D matrix of size L × L.

As mentioned in the Introduction section, a typical pipeline for 3D protein structure prediction consists of contact map prediction: predicting residue–residue distance relationships (e.g., contacts) has become the key direction to advance protein structure prediction since the CASP11 experiment. Most recently, Deep Learning has revolutionized the technology for contact and distance prediction since its debut in the 2012 CASP10 experiment.

Protein contact map prediction has been deemed to be useful in predicting protein structure for more than a quarter of a century [33]. Protein contact prediction can be broadly classified into two categories [34]: correlated mutation-based methods and machine learning (ML)-based methods. For a review on protein residue contacts and prediction methods and ML methods in contact map prediction, please refer to [34] and [35], respectively. Adhikari and Cheng [34] summarize the protein residue contact map problem and outline some of the machine learning-based approaches for protein contact map prediction. Xie et al. [35] provide details of neural network-based methods as well as other machine learning-based methods for protein contact map prediction.

One of the earliest DL-based approaches for protein contact map prediction is DNcon [36] developed by Cheng’s group, which used Restricted Boltzmann machines (RBMs) trained to form Deep Neural Networks. It was the first method to use Deep Learning, and it was ranked at the top in the CASP11 competition in the contact prediction category. Soon after, the convolutional neural network-based method was introduced by Xu’s group, which demonstrated much higher accuracy [8].

After that, as contact map established itself as an intermediate step in most of the successful protein prediction pipelines, a lot of advances have been made in contact prediction using Deep Learning-based approaches. To summarize, some of the earlier Deep Learning-based contact map predictors are DeepCDPred [37], RaptorX-Contact [9], DNCON2 [10], PconsC4 [38], and SPOT-contact [39]. Readers are suggested to refer to either Torissi et al. [23] for background and earlier Deep Learning methods for protein contact map prediction. Here, we review some of the more recent Deep Learning-based approaches.

#### 2.2.1. RaptorX-Contact

RaptorX-Contact [8] is a contact map prediction tool developed by Xu’s group based on Deep Learning. RaptorX-contact uses deep layers and consists of two major residual neural network modules. The first module conducts a series of 1D convolutional transformations of sequential features (sequence profile, predicted secondary structure, and solvent accessibility). The output of the 1D network is converted to a 2D matrix and then fed into the 2nd module combined with other pairwise features (co-evolution information, pairwise contact etc.). The 2nd module is a 2D Residual Neural Network (ResNet) [40] module that conducts a series of 2D convolutions. Finally, the output of 2D convolutional network is fed into logistic regression. This technique of using 1D convolutions to learn 1D features and 2D convolutions to learn 2D features is an elegant way to learn feature representations and save computational resources. When tested on CASP11 targets, RaptorX-Contact produced better results compared to other existing approaches such as MetaPSICOV [41] and CCMPred [42]. One of the salient features of RaptorX-contact is that it is one of the intermediate steps of one of the most successful protein structure prediction pipeline, RaptorX. The recent version of RaptorX-Contact [43] also predicts distance map. This method was ranked top in CASP12 and CASP13 in the contact prediction category.

#### 2.2.2. ResPre

Zhang’s group developed a deep residual convolutional neural network-based approach called ResPre [44] to predict residue–residue contacts multiple sequence alignment. Initially, for a query sequence, MSA is created. Based on the MSA, the covariance matrix is created, and this covariance matrix is converted to a precision matrix, which is then fed to the residual convolutional neural network. It has to be noted here that the quality of MSA will be critical for the eventual contact map prediction, and the authors also paid attention to create deep MSA. One of the novelties of this work was the use of precision matrix.

#### 2.2.3. MapPred

Yang’s group developed MapPred [45], which uses a metagenome sequence in a residual neural network framework. MapPred uses the vast amount of data from metagenome sequencing projects to overcome limitations in protein sequence databases.

For each query sequence, MapPred first generates an MSA and then feeds the covariance features derived from MSA into a deep residual neural network. Essentially, a contact map using CCMPred [42] and DeepMSA is constructed from the MSA and then Position-specific scoring matrix (PSSM) is also created. In addition, the secondary structure and relative solvent accessibility of the sequence are predicted using PSIPRED [46], and these features are transformed into 2D by pairing. Finally, all the features are concatenated, and the final contact map is predicted. One of the observations of this approach is that the contribution from the metagenome sequence is statistically significant. In addition to contact map, MapPred also predicts distance maps and the distance distribution.

#### 2.2.4. DEEPCON

DEEPCON [47] discusses how deep convolutional neural network methods (ConvNets) may be best designed and developed to solve the distance prediction problem. With publicly available datasets, the work designed and trained various ConvNet architectures, including wide residual networks, dropouts, and dilated convolutions. In this work, they have studied several deep learning architectures to improve the precision of medium-range and long-range contact. Furthermore, they have highlighted the performance comparison of the recently developed state-of-the-art methods by comparing them with their best method. The ConvNet architectures discussed in this work—ConvNets with alternating dilations and dropout—predict contacts with significantly more precision than the architectures used in several state-of-the-art methods. They have reported that there was a 15% improvement in the top L/2 long-range contacts precision on the 150 proteins PSICOV test datasets when the network was trained with 3456 proteins from the DeepCov dataset. The work also discusses how popular architectures such as U-net are inappropriate for distance prediction.

#### 2.2.5. DeepECA

Fukuda et al. developed DeepECA [48], a Convolutional Neural Network (CNN)-based approach on evolutionary coupling analysis, to predict contact map directly from MSA in an end-to-end manner. Based on CASP results and other benchmarks, this approach can use information derived from either deep or shallow MSAs. In addition, using multi-task model to predict secondary structure and contact simultaneously, DeepECA shows some improvement in the secondary structure prediction.

#### 2.2.6. ContactGAN

Kihara’s group recently developed a Generative Adversarial Network (GAN) [49] based approach for contact map prediction called ContactGAN [50]. This approach is quite novel compared to the existing contact map prediction approaches as the task here is to refine the predicted contact map.

As ContactGAN is a contact map refining tool, the input is a predicted contact map (rather than an MSA) and the output is a refined contact map. Essentially, the DL architecture of ContactGAN uses a GAN framework, where generative and discriminative networks are trained with sets of predicted (noisy) and corresponding native (correct) contact maps. The generator network takes a noisy predicted contact map and outputs a refined map, whereas the discriminator network discriminates a generated map and the native map, so that the generator is trained to produce indistinguishable maps from native maps by the discriminator. The generator network is based on ResNet blocks and the Discriminator network is based on CNN. ContactGAN when applied to predict contact maps of CCMPred [42], DeepCov [51], and DeepContact [52] a consistent improvement in the protein contact map was obtained.

#### 2.2.7. InterPretContactMap

As in other fields, the black-box nature of deep learning models has been one of the major hesitation/roadblocks for more widespread implementation/usage of DL-based approaches in Protein Structural bioinformatics. In that regard, the new trend in the field is to develop xAI (explainable Artificial Intelligence) approaches. InterPretContactMap [53] is one of the approaches toward xAI. Combining deep neural network with attention mechanisms to enhance the explainability of protein contact prediction is one of the new approaches in contact map prediction that gears toward the explainability of the predicted contacts. Using two attention mechanisms (sequence and regional), in the CNN framework for contact map prediction, InterPretContactMap improves the contact map prediction results as well as provides some level of interpretability, providing some insights into the key fold-determining residues in proteins.

#### 2.2.8. TripletRes

TripletRes [54], also developed by Zhang’s group, is another DL-based approach to predict protein contact maps. TripletRes consists of three steps: deep multiple-sequence alignment generation, co-evolutionary feature extraction and Deep Neural Network modeling. One of the important steps in TripleRes is the construction of deep MSA, which is obtained by applying multiple iterations of HHblits [30]. Second, three sets of co-evolutionary features viz. covariance features (COV), precision matrix features (PRE), and a coupling parameter matrix approximated by pseudolikelihood maximization (PLM) are extracted from the deep MSAs created in step 1—hence the name TripletRes. Finally, these features are fed to a residual neural network. 

The TripletRes contact pipeline performed quite well in two independent sets of test targets that included 50 non-redundant Free Modeling (FM) targets from CASP11 and CASP12. The major advantage of TripletRes is its ability to learn and directly fuse a triplet of co-evolutionary matrices extracted from the whole-genome and metagenome databases and therefore minimize the information loss during the course of contact model training. TripletRes also achieved the highest precision (71.6%) for the top-L/5 long-range contact predictions in the CASP13 experiment in the contact prediction category.

#### 2.2.9. Summary of Advances in DL-Based Approaches for Protein Contact Map Prediction

Overall, a wide range of features such as precision matrix, covariance matrix, secondary structure, PSSM, sequence profiles, and solvent accessibility have been proven to be important for protein contact map prediction as observed in all contact prediction methods. Similarly, a variety of Deep Learning architectures including ResNet, fully convolutional neural networks (FCNs), GANs, and U-Nets have been implemented by different groups. Among those architectures, we find that all accurate methods use ResNet and its variants. CASP-winning methods such as TripletRes [54] and RaptorX [8] serve as examples. All of these recent methods for contact prediction also discuss that the use of deeper multiple sequence alignments (MSAs) as inputs is the key for accurate predictions. However, after the CASP13 competition, it was evident that deep learning methods could predict more richer information than mere binary contacts. Hence, most groups switched from contact prediction to distogram prediction or real-valued distance prediction as they provide much richer information for building models.

### 2.3. Deep Learning-Based Advances in ‘Distogram Prediction’

Inter-residue contact prediction has been widely used for over a decade in the field of protein structure prediction. However, recently, the paradigm shifted toward predicting the probability of distance intervals, also known as ‘distograms’ [12]. After the Xu group and DeepMind’s AlphaFold demonstrated that predicting distograms can be more informative than their binary counterparts (contacts) [55], many groups pursued similar approaches. Notable methods such as trRosetta [56], the ResNet/Densenet-based method [57], and DeepDist [16] showed results similar to or better than the top groups in the CASP13 competition.

The protein inter-residue distance prediction is the prediction of a pairwise distance matrix (2D) from a protein sequence (1D amino acid sequence). It can be compared with the monocular or stereo depth estimation problem in computer vision as shown in Figure 3. In image depth prediction, an image matrix is provided as input and a depth matrix is predicted as an output where each pixel has a predicted depth (distance from the camera to the object). Similar to the depth prediction problem, the distance prediction takes a three-dimensional input volume (height × width × channels) and outputs a distance map with the same dimension as the input (height × width) but with a single channel. However, the input channels in computer vision problems range from one to three, but there are a few to few hundred channels in distance prediction problems depending on the input features. Additionally, the distance map is symmetrical about the diagonal, and each pixel on the map represents a distance between a pair of residues in the sequence.

Although there were a few previous studies [58,59] on predicting protein structure based on protein distogram/distance, until AlphaFold [12] and Xu’s method [11], those methods were not quite satisfactory. In that regard, AlphaFold, Xu’s method, and trRosetta [56] can be attributed the success of using distogram prediction in the protein structure prediction pipeline.

#### 2.3.1. Distogram Prediction in Xu’s Approach

Xu’s group during CASP13 implemented ‘distogram prediction’ [11] within their RaptorX pipeline. This method predicts the ‘distogram’ using a ResNet architecture that consists of one 1D deep ResNet, one 2D deep dilated ResNet, and a softmax layer. Essentially, discretizing the interatom distance Cβ-Cβ into 25 bins (<4.5 Å, 4.5 to 5 Å, 5 to 5.5 Å, …, 15 to 15.5 Å, 15.5 to 16 Å, and >16 Å) and treating each bin as a classification label, this DL model predicts the distance matrix. It has to be noted here that a contact map can be obtained from this distance map by summing the predicted probability values corresponding to distance ≤8 Å. Finally, the RaptorX framework predicts the protein structure using this predicted interatom distance matrix, secondary structure, and backbone torsion angles using the Crystallography and NMR (CNS) [60] framework. The prediction accuracy of RaptorX using the distance matrix was quite better than that using contact matrix on a set of CASP targets. Along with AlphaFold, this approach helped establish the fact that protein distogram can be predicted quite well and that predicted distogram is better than the contact map for protein structure prediction.

#### 2.3.2. Distogram Prediction in AlphaFold

AlphaFold [12] is a protein structure prediction method developed by DeepMind, which had one of the best performances in CASP13. AlphaFold is the other approach that championed the idea of using distogram for protein structure prediction. Essentially, the distogram prediction component of AlphaFold uses a convolutional neural network (CNN) that is trained on PDB structures to predict the C_β_–C_β_ distances between any pair of residues. Using the amino acid representation of the query sequence and features generated from MSA, the CNN network predicts a discrete probability distribution for every pair. This distribution is found to be similar to the true distances. Then, the predicted backbone torsion angles and pair-wise distance between residues are combined to form a protein-specific potential. Finally, the gradient descent on protein-specific potential is applied to obtain the final protein model.

#### 2.3.3. ProSPr

ProSPr [15] is an unofficial and democratized (open-source) implementation of AlphaFold protein distance and structure prediction method. In this work, they have re-implemented the Deep Learning part of AlphaFold for intramolecular distance prediction and made the source code freely available. The Deep Learning method uses a residual network with 220 residual blocks with batch normalization followed by exponential linear unit (ELU) activation function, and a cycle of four different dilation filters of sizes 1, 2, 4, and 8, respectively. The output of the network consists of distance and auxiliary predictions where the auxiliaries predict eight classes of secondary structures as defined with Dictionary of Secondary Structure of Proteins (DSSP) classifications. They tested their method on the CASP13 dataset for a free and template-based model. They convert the distance probabilities into contact probabilities and calculate the precision score. The precision scores of their method are comparable to the winning group TripletRes [54]. ProSPr is 2% better than TripletRes on average on the L/5 scores for the high confidence prediction (with maximum <8 Å).

#### 2.3.4. Distogram Prediction in trRosetta

Another seminal work that championed the use of distogram is trRosetta (transform-restrained Rosetta) [56], which is a newer Rosetta-based method developed by Yang’s group and Baker’s group. Similar to AlphaFold and Xu’s work, the distogram prediction component within trRosetta takes MSA as an input and then uses a deep residual–convolutional neural network (stack of dilated residual–convolutional blocks) to predict the relative distances (distance prediction) along with orientations of all residue pairs in the protein. The features used in the prediction include one-hot-encoded amino acid sequence (20 features), position-specific frequency matrix (21 features) and positional entropy (1 features). The distance range (2 to 20 Å) is binned into 36 equal segments of 0.5 Å each. The last convolutional layer is followed by a softmax function that predicts the probability for each of these bins given a residue pair. Subsequently, the protein structure model is built based on restrained minimization using predicted distance and orientation restrains as in AlphaFold [12]. One of the novel ideas in trRosetta is the prediction of inter-residue geometry/orientations where orientations between two residues are represented by three dihedral and two planar angles between the residues.

#### 2.3.5. AttentiveDist

Kihara’s group developed a DL-based method called AttentiveDist [61] for protein inter-residue distance prediction. The network is derived from residual network (ResNets) with an added attention mechanism to determine the most relevant MSA for each residue pair. There are 45 residual blocks in the network where the first five blocks are used for feature encoding. Four different MSAs of different E-values are used to generate four different inputs for the network to train. At the end of the last residual block, the model branches out into five different paths to predict five different outputs, namely, Cβ distance prediction, three side-chain orientation angles for each amino acid residue pair, and backbone dihedral angles. The success of AttentiveDist shows that the use of an attention-based approach on different MSA-based features is correlated to the co-evolutionary information in the MSA. AttentiveDist has an average TM-score of 0.579 on 43 CASP13 targets, which is slightly better than the top CASP13 server model, which has an average TM-score of 0.517.

#### 2.3.6. Summary of Deep Learning-Based Advances in ‘Distogram Prediction’

All distogram prediction methods use ResNet as the core deep learning architecture. The RaptorX method [11] was ranked as the top method for distance prediction in the CASP13 competition. Since the AlphaFold method did not participate in the contact prediction category of the CASP13 competition, it is difficult to evaluate the distograms generated by AlphaFold. The trRosetta method [56] and RealDist method [17] (discussed in the next section) developed after the CASP13 competition are shown to outperform the top CASP13 predictor. Both trRosetta and Alphafold predict inter-residue orientations, which can help to improve the structure prediction. Although all of these methods are publicly available, we found trRosetta to be the easiest to use because it is a fully TensorFlow-based implementation and does not require any feature generation after MSAs are ready.

### 2.4. Deep Learning-Based Advances in ‘Real-Valued Distance Prediction’

An ideal distance prediction algorithm should predict exact physical distances on the entire distance map accurately [18]. Although this is extremely difficult, many groups focused on predicting real-valued distances i.e., predicting what the distances truly are in nature. Real-valued distance prediction has many advantages over contact prediction. First and foremost, a predicted distance map has much more detailed information about the structure than contact prediction, which is just human-defined zero-one labels (binary classification), whereas distance map is a real-valued physical metric. Secondly, because of the binary nature of contact prediction, it causes imbalance in positive and negative samples; i.e., there will be one contact to 50 non-contacts for a long-range residue pair with sequence separation ≥24, which leads to undersampling of negative ones during model training. As a result, there may be inconsistency between the real contacting probability for a residue pair and prediction score [18]. Below, we summarize some of the successful Deep Learning methods for real-valued distance prediction.

#### 2.4.1. PDNET

In this work, the author has developed a framework called Protein Distance Net (PDNET) [19] that consists of a small and representative protein dataset which can be used for faster training and testing of Deep Learning methods for distance prediction. This is one of the first methods that was made public to the scientific community as a protein distance map prediction framework. A variant of residual network (ResNet) was used in this work as the Deep Learning architecture for the training. They developed three separate Deep Learning methods to predict contacts (PDNET-Contact), distance intervals or distograms (PDNET-Binned), and real-valued distance (PDNET-Distance), respectively. The Deep Learning architecture uses 128 residual blocks with dropouts added in between the convolutional layers, which is described properly in DEEPCON [47] method. PDNET uses a unique in-house loss function in its approach, which focuses on predicting shorter distances more precisely than longer distances because from the perspective of structure prediction and binding-site prediction, it is more meaningful to predict inter-residue interactions than non-interactions.

#### 2.4.2. GAN-Based Real-Valued Distance Prediction Method

Ding et al. [18] developed a new GAN-based Deep Learning method to predict real-valued inter-residue distances. The method was developed by adding GAN on top of the residual network (ResNet) to enforce global distance consistency. The method was trained mainly by protoplasmic soluble proteins. The approach adopts a conditional GAN (cGAN) where 40 layers of ResNet were used as the generator, which generates the output for the discriminator of cGAN that was trained to detect the output as fake or real. Then, the decision of the discriminator was used by the generator to learn and produce outputs to fool the discriminator through an adversarial training procedure. In this way, the model was trained to predict real-valued distances. This method produced structural models with real-valued distance-based structure prediction with an average TM score of 0.620 and 0.786 for the FM targets and Template-Based Modelling (TBM) targets, respectively, which are comparable with the top CASP13 groups. In addition, the method has an average TM score of 0.712 for all available 42 CASP13 targets, which is higher than the top CASP13 groups.

#### 2.4.3. Xu’s Real-Valued Distance Prediction Method

Xu’s group also developed a Deep Learning-based method [62] to predict real-valued distances as well as inter-residue orientations. A deep residual network (ResNet) with 60 residual blocks which is described in [43] with detailed information has been used in this work. In addition to predicting real value distance, the method also predicts the mean and standard deviation of a distance. The predicted mean and standard deviations for building the 3D structure models were built using PyRosetta [63]. To build the model, they trained six deep ResNet models of the same architecture on the same training data in order to predict real-valued and multi-class distances using ensemble methods. Validation results show that the real-value distance prediction obtains 81% precision on top L/5 long-range contact prediction, which is better than the best CASP13 results (70%), and it predicts folds for the CASP13 FM targets as correctly as the best group in CASP13, outperforming the DeepDist method mentioned above.

#### 2.4.4. RealDist

RealDist [17] is a purely real-valued distance map prediction method, i.e., the method only predicts distance map and does not predict distograms or orientations. For training the ResNet-based model, a set of 43,000 protein chains was used—the largest dataset ever used for training distance prediction methods. The network architecture is very deep and consists of 256+ convolutional layers. The contacts derived from the real-valued distance maps predicted by this method, on the most difficult CASP13 free-modeling protein datasets, demonstrate a long-range top-L precision of 52%, which is 17% higher than the top CASP13 predictor Raptor-X and slightly higher than the more recent trRosetta method. Similar improvements were observed on the CAMEO ‘hard’ and ‘very hard’ datasets. 3D structure prediction guided by real-valued distances showed that for short proteins, the mean accuracy of the 3D models is slightly higher than the top human predictor AlphaFold and server predictor Quark [64] in the CASP13 competition.

#### 2.4.5. DeepDist

Cheng’s group developed DeepDist [16], which is a DL-based method where residual convolution network architectures are used to simultaneously predict real-value inter-residue distances as well as classify them into multiple distance intervals (such as bin classification used in trRosetta and AlphaFold1). Validation results for DeepDist show that predicting a real-value distance map has some added value on top of predicting a multi-class distance map. When DeepDist was used to build 3D models for the 43 CASP13 hard domains, the obtained average TM-scores of the top model and the best model of the top five models were of 0.487 and 0.522, respectively. In addition, in the comparison study where the models were built from real-value distance prediction and multi-class distance prediction, it was found that real-value predictions have higher scores.

#### 2.4.6. DISTEVAL

Contact and distance predictions have been widely used as the intermediate steps toward protein structure prediction. The quality of the 3D structure prediction depends on the accuracy of such contact and distance predictions, which has been shown in the recent 13th and 14th CASP experiments. Therefore, it is important to assess and evaluate such predicted contacts and distances. To evaluate predicted contacts, web servers such as EVAcon [65] and ConEva [66] are freely accessible. However, there are currently no methods to evaluate predicted distograms and real-valued distance maps. DISTEVAL was developed to fill this void. DISTEVAL can evaluate predicted contacts, multi-class distance prediction (distograms), as well as real-value distance predictions. It performs both qualitative assessments using heatmaps, chord diagram, and 3D model visualization (if true structure is provided) as well as quantitative assessment using metrics such as mean absolute error (MAE) of long-range distance, root mean square error (RMSE), local distance difference test (lDDT) score, and precision of medium and long-range contacts. All features offered by DISTEVAL collectively serve as a powerful tool to compare and assess predicted contacts, distograms, and distances even in the absence of a true 3D structure. Since methods such as AlphaFold2 still use distance prediction (distogram) in their end-to-end pipeline for interpretation while predicting protein structure, DISTEVAL could be used further to investigate what these deep learning models are learning.

#### 2.4.7. Summary of Deep Learning-Based Advances in ‘Real-Valued Distance Prediction’

The methods discussed above represent the recent developments in real-valued protein distance prediction. While most of them use deep residual networks (ResNets), methods based on generative adversarial networks (GANs) and Attention Networks have also been proposed. The results of the methods such as REALDIST [17], DeepDist [16], and the Xu group’s work [62] show that real-valued distance prediction is as promising as distogram prediction. These methods propose various novel and complementary methods to predict real-valued distances, suggesting that merging these ideas would lead to more accurate real-valued distance predictions.

### 2.5. DL-Based Advances in Ranking of Models, Quality Assessment (QA), and Refinement

Ranking of models, Quality Assessment (QA), and refinement are some of the integral steps in the protein structure prediction pipeline. For example, in I-TASSER [26], once the models are generated using Replica Exchange Monte-Carlo Simulation, the best model is obtained by performing clustering of the decoys and subsequent refinement. In recent years, there have been significant improvements in the performance of estimating model accuracy (EMA) algorithms partly due to the application of Machine Learning, and EMA methods based on ML have been consistently ranked among better predictors. Please see Chen and Siu [67] for details about machine learning approaches (and a few Deep Learning approaches) published until 2019 for quality assessment of protein structures. Here, we summarize some recent DL-based advances for QA and EMA. In addition, QA methods can be divided into two types: consensus methods (multi-model) and single-model methods. Mainly, consensus methods depend on comparison of models of a protein target and are performed when a protein has many models generated by different predictors and a single model depends on predicting quality of a model using only its own information.

#### 2.5.1. QDeep

Bhattacharya’s group recently developed a distance-based model quality estimation method called QDeep [68]. One of the salient features of this approach is that the model is trained on an ensemble of stacked deep ResNets that can perform residue-level error classification at multiple error thresholds. Finally, the individual error classifiers are then combined to estimate the quality of a protein model. Essentially, QDeep consists of four steps: (i) multiple sequence alignment generation, (ii) feature extraction from distance-based alignment, sequence and structure, and ROSETTA centroid energy-terms, (iii) residue level classifiers at 1, 2, 4, and 8 Å error thresholds, and (iv) ensemble error classifier for protein model. Benchmarking results on CASP12 and CASP13 targets showed that QDeep performs pretty well in comparison to other state-of-the-art algorithms. One of the novel features in this approach is the use of distance-based features. This is also one of the DL-based approaches where a large ResNet model [40] is used as the DL architecture for protein EMA.

#### 2.5.2. ResNetQA

Xu’s group developed ResNetQA [69] (a ResNet-based QA method for both local and global quality assessment of a protein model. ResNetQA is a single-model method. In ResNetQA, an MSA is built using HHblits, and then, three types of features viz. sequential, co-evolution, and predicted distance potentials are passed to the Deep Neural Network (DNN). The sequential feature includes the one-hot-encoding of primary sequence, PSSM matrix derived from MSA, three-state secondary structure, and solvent accessibility. The co-evolutionary based feature included the output of CCMPred, and the distance potential is obtained from distance distribution predicted by RaptorX-Contact. Then, these features are passed to a Deep Model that consists of 1D and 2D dilated residual neural network based on ResNet. The final ResNetQA model has 21 2D convolutional layers and 16 1D convolutional layers. ResNetQA predicts a residue-wise S-score for local QA. The ResNetQA model when tested on CASP12 and CASP13 datasets performed better than methods such as QDeep [68].

#### 2.5.3. MULTICOM EMA Predictors

Similarly, Cheng’s group developed six EMA predictors (MULTICOM) using Deep Learning [70] with features from inter-residue distance/contact prediction, other existing single-model features, and multi-model quality features. Given a protein target sequence and a pool of structural models for the target, a MULTICOM EMA predictor first invokes an inter-residue distance predictor (DeepDist [16]) and or/an inter-residue contact predictor (DNCON2 [10]). Finally, several distance/contact-based features and other non-distance/contact features used in DeepRank [71] are generated, and different combinations of features are used in a Deep Learning framework to predict the Global Distance Test—Total Score (GDT-TS) score of a model. Among these six EMA predictors, based on the convention in the field, MULTICOM-CLUSTER, MULTICOM-CONSTRUCT, MULTICOM-AI, and MULTICOM-HYBRID can be classified as multi-model methods and MULTICOM-DEEP and MULTICOM-DIST can be classified as single model methods based on whether the features are based on comparison between multiple models as input or just from a single model, respectively. The MULTICOM EMA family of methods performed quite well in CASP14; especially, MULTICOM-CONSTRUCT had a GDT-TS Loss of 0.07356 and was ranked among the top among all the methods for EMA in CASP14.

#### 2.5.4. DeepAccNEt

Baker’s group recently developed a DL framework called DeepAccNEt [72] that estimates the error in every residue–residue distance along with the local residue contact error. DeepAccNEt consists of a series of 3D and 2D convolutional layers and predicts (i) error histogram (Cβ–Cβ distance error distribution), (ii) mask (native Cβ contact map with a threshold of 15 Å, (iii) and per residue Cβ local distance difference score (Cβ I-DDT) score. The input features to the networks are distance maps, amino acid identities, local atomic environments scanned with 3D convolutions, backbone angles, residue angular orientations, Rosetta energy terms, and secondary structure information. In addition, MSA information in the form of inter-residue distance prediction by the trRosetta and sequence embeddings from the ProtBert-BFD100 model are also optionally provided as 2D features. DeepAccNEt incorporates 1D, 2D, and 3D features. Initially, the network performs a 3D convolution operation on local atomic grids, and as a result, features describing the local 3D environments of each of the N residues are generated. The 1D features (local torsional angles and individual residue energies) are also combined with the 2D residue–residue input features, and the resulting combined 2D feature is input to a series of 2D convolutional layers using the ResNet architecture. DeepAccNEt performed well in comparison to other methods, and the incorporation of DeepAccNEt in Rosetta refinement protocol helped achieve better results.

#### 2.5.5. Summary

The necessity for mitigating error and increasing the accuracy of the predicted 3D protein structure has led to the development of methods for the refinement of 3D models. The methods discussed here represent the most recent developments in the field of quality assessment in the estimation of model accuracy (EMA). QDeep [68] uses ResNet for estimating the quality of a protein structural model. Similarly, ResNetQA [69] is a local and global quality assessment method composed of both 1D and 2D convolutional residual neural networks (ResNet). DeepAccNet [72] estimates per-residue accuracy and residue–residue distance signed error in protein models and uses these predictions to guide Rosetta protein structure refinement using a convolutional neural network. 

## 3. Deep Learning-Based Advances in Overall Protein Structure Prediction Pipelines

In this section, we highlight the recent advances in overall protein structure prediction pipeline using Deep Learning. Mainly, we will discuss recent Deep Learning-based improvements in some of the most successful protein structure prediction pipelines as evidenced by CASP13 and CASP14 and some end-to-end DL-based approaches for protein structure prediction.

### 3.1. DL-Based Advances in Protein Structure Prediction Pipeline

Here, we highlight DL-based advances in some of the most successful protein structure prediction pipelines.

#### 3.1.1. AlphaFold

AlphaFold [12] is a protein structure prediction method developed by DeepMind, which had one of the best performances in CASP13. AlphaFold works on the premise that given a protein sequence, it is possible to construct a learned, protein-specific potential by training a Deep Neural Network (DNN) to make accurate predictions about the structure and to predict the structure itself by minimizing the potential by gradient descent. The features used in the DNN are MSA features generated by running HHblits and PSI-BLAST [73] on sequence databases. The DNN predicts backbone torsion angles and pair-wise distance between residues. Then, the predicted distance and torsion probability distributions along with Van Der Walls are combined to form a protein-specific potential. Finally, gradient descent on protein-specific potential is performed to obtain the final protein model. The training data for the model is extracted from PDB and more specifically CATH domains where 29,427 proteins were used for training and 1820 proteins are used for testing. The good performance of AlphaFold is attributed to the accuracy of distance predictions. The most recent version of AlphaFold which is termed as AlphaFold2 produces even remarkable results, and it will be interesting to see the overall aspects of AlpahFold2. The notion of minimizing the potential by gradient descent rather than using Fragment Assembly and subsequent model refining is quite novel.

#### 3.1.2. trRosetta

Similar to AlphaFold, trRosetta (transform-restrained Rosetta) [45] is a new Rosetta-based method for protein structure prediction given a protein sequence developed by Ying’s group and David Baker’s group. Essentially, trRosetta takes the amino acid sequence of a query protein and then computes MSA from the input. Taking MSA and homologous templates obtained used HHsearch, the combined features are then passed to a ResNet-based deep residual–convolutional neural network to predict the relative distances (distance prediction) and orientations (represented by maps of φ (phi), ω (omega), and θ (theta), of all residue pairs in the protein. Subsequently, the protein structure model is built based on restrained/constrained minimization using predicted distance and orientation restraints as in AlphaFold [12]. This is also quite unique in that the fragment assembly approach was not utilized as in original Rosetta; rather, folded structures satisfying the restraints were generated starting from conformations with randomly selected backbone dihedral angles. One of the novel ideas in trRosetta is the prediction of inter-residue geometry/orientations where orientations between two residues are represented by three dihedral and two planar angles between the residues.

Using similar ideas as trRosetta, Gray’s group expanded the ideas to predict antibody structure prediction and developed DeepH3 [74], which is a deep residual network-based model that predicts inter-residue distance and orientations from antibody heavy and light chain sequence. Subsequently, these distributions are converted to geometric potentials and used to discriminate between decoy structures produced by RosettaAntibody and predict new CDR H3 loop structures de novo. DeepH3 did better than the vanilla Rosetta energy function model as well as inter-residue orientations were more effective than inter-residue distances for discriminating near-native H3 loops.

#### 3.1.3. RaptorX

RaptorX is also one of the successful protein structure prediction pipelines developed by Xu’s group. The most recent version of RaptorX [43] also incorporated the notion of distance prediction. The Deep Learning network used in RaptorX to predict distance matrix consists of one 1D Deep ResNet, one 2D deep dilated ResNet, and one softmax layer. Xu’s distance prediction approach is also described briefly in Section 2.3 and Section 2.4. Given a protein sequence, initially, RaptorX-Contact predicts the predicted interatom distance matrix, secondary structure (three-state), and backbone torsion angles, and then, these predictions are converted into CNS restraints for 3D model building using CNS [60]. Finally, for each protein, 200 possibly decoys are generated by CNS, and then then models with the least violation of distance restraints are chosen as the final models. The prediction accuracy of Raptor-X contact predicts the interatom distance matrix, secondary structures using the distance matrix was quite better than that using contact matrix on a set of CASP targets. RaptorX was successful in folding 17 of 32 hard targets in CASP13.

#### 3.1.4. MULTICOM

Cheng’s group, the primary developer of MULTICOM (another protein structure prediction pipeline), has also incorporated recent advances in DL-based approaches to improve the MULTICOM [75] protein structure prediction system. The recent version of MULTICOM protein structure prediction pipeline added three main improvements: (a) a new Deep Learning-based protein inter-residue distance predictor DeepDist [16] to predict protein inter-residue distance, (b) an enhanced template-based tertiary structure prediction method, and (c) a DL-based framework for assessing model quality using predicted residue distance.

Given a target protein sequence, initially, for the template-free modeling, MULTICOM generates MSA and then the MSA is used to calculate residue–residue co-evolution features that are passed to the DNN of DeepDist to predict the inter-residue distance map. Then, the MSA and predicted distance map are used to generate ab initio models tools (e.g., trRosetta). Finally, MULTICOM EMA predictors are used to rank the models. The new MULTICOM was ranked 7th out of all the predictors in protein 3D structure prediction.

#### 3.1.5. C-QUARK/C-I-TASSER

Yang Zhang’s group also has been incorporating DL-based advances in the I-TASSER pipeline. Especially, during CASP13, the group released C-I-TASSER and C-QUARK programs [64], based on I-TASSER and QUARK, C meaning contact. The new incorporations in I-TASSER pipelines are (1) the incorporation of DeepMSA [20], a novel multiple sequence alignment (MSA) generation protocol to construct deep sequence profiles for contact prediction; (2) an improved meta method, NeBcon, which combines multiple contact predictors, including ResPRE that predicts contact maps by coupling precision matrices with deep residual convolutional neural networks; and (3) an optimized contact potential to guide structure assembly simulations.

Essentially, as in previous iterations of I-TASSER, the C-ITASSER pipeline consists of the following steps. (a) Given a protein sequence, the sequence is threaded using LOMETS, and at the same time, MSA is generated using DeepMSA. (b) Template fragments are created from the threading templates, which are then subjected to structure assembly using Replica-Exchange Monte Carlo (REMC) guided by the potential calculated from the improved contact map created using NeBcon (that incorporates other DL-based contact prediction approaches). Finally, the models are clustered using SPICKER, and the cluster centroid is chosen and subjected to structure re-assembly and finally, Fragment-Guided Molecular Dynamics (FG-MD)-based refinement is applied to obtain the final model.

In CASP13, the average TM scores of the first models produced by C-I-TASSER and C-QUARK were 28% and 56% higher than those constructed by I-TASSER and QUARK, respectively. Detailed data analyses showed that the success of C-I-TASSER and C-QUARK was mainly due to the increased accuracy of Deep Learning-based contact maps, as well as the careful balance between sequence-based contact restraints, threading templates, and generic knowledge-based potentials. In CASP14, I-TASSER also incorporated a distance matrix, but the details are yet to be released.

#### 3.1.6. Summary

There have been some advances in some of the most successful protein structure prediction pipelines such as RaptorX, I-TASSER, Rosetta, MULTICOM, and AlphaFold. These advances can be mostly attributed to the improvement in one or more stages of the prediction pipeline due to the use of DL-based approaches. Especially, for all of these pipelines, these advancements have been integrated into the original pipeline and are available as web servers or a standalone version of the tools provided in Github.

### 3.2. Advances in End-To-End Deep Learning-Based Approaches for Protein Structure Prediction

As in other scientific fields, there have been some attempts to get rid of human-engineered steps in the protein structure prediction pipeline. It has to be noted here that the existing approaches contained a number of modules, which were each trained separately even though they were using DL, whereas an end-to-end deep learning system would be trained as a single integrated structure with a system of sub-networks coupled together. In that regard, end-to-end DL approaches use sequence as an input and map to protein structures end-to-end. In this section, we review a few end-to-end DL-based approaches for protein structure prediction.

#### 3.2.1. NEMO

NEMO [76], which stands for Neural Energy Modeling and Optimization, is one of the first end-to-end differentiable DL-based approaches for protein structure prediction. NEMO takes the protein sequence as an input and generates a 3D protein structure directly from sequence information. NEMO consists of three components: (i) a neural energy function for a coarse-grained structure given sequence, (ii) an unrolled simulator that generates an approximate sample from energy function, and (iii) an imputation network that generates an atomic model from the final coarse-grained sample. It has to be noted that all these components are trained simultaneously via backpropagation. Protein sequences are represented/conditioned using one-hot encoding and a profile of evolutionary related sequences. Since NEMO has not participated in CASP competition, its comparative performance is not quite known. However, it is one of the seminal works for the end-to-end Deep Learning approach for protein structure prediction.

#### 3.2.2. AlQuraishi’s Recurrent Geometric Network

Another seminal work for the end-to-end Deep Learning approach for protein structure prediction is based on AlQuraishi’s work [14]. Based on the idea that end-to-end differentiable DL has revolutionized various fields such as computer vision and speech recognition, AlQuraishi attempted to get rid of many human-engineered steps/stages in the protein structure prediction. Essentially, acknowledging the fact that the advances in DL-based approaches so far were mainly focused on each step of the protein structure prediction pipeline (please see Figure 1) such as contact map prediction, distance map prediction, QA, etc., AlQuraishi proposed an end-to-end differentiable model to predict the overall structure pipeline. The model termed as recurrent geometric network (RGN) predicts the structure of one segment of a protein partly on the basis of what comes before and next. The framework that takes the amino acid of the target sequence as input and outputs a 3D structure of the target sequence is based on four core ideas: (i) encoding protein sequence using a recurrent neural network, (ii) parameterizing (local) protein structure by torsional angles, (iii) coupling the local protein structure to its global counterpart using recurrent geometrical units, and (iv) capture deviations between predicted and native structures using a loss function. Essentially, the model predicts the most likely angle of the chemical bonds that connect the amino acid with its neighbors for each amino acid. It also predicts the angle of rotation around these peptide bonds. This process is repeated where each calculation is informed and refined by the relative positions of other amino acids. Finally, the model accuracy is checked by comparing it to the native structure of the protein. The approach is trained using known PDB structures. The approach was able to build with reasonable accuracy novel folds as well as predict known folds without templates. Although rigorous benchmarking still needs to be done, the approach seems to be much faster for prediction compared to traditional approaches. It will not be an overstatement to say that this approach probably inspired AlphaFold2.

#### 3.2.3. AlphaFold2

Although the details of AlphaFold2 (recent version of AlphaFold [12]) have not yet been released, some aspects of AlphaFold2 have been made clear during the CASP14 meeting. AlphaFold2 (AF2) is an end-to-end deep learning-based method trained using a dataset of around 170,000 proteins (which is almost the entire Protein Data Bank). One key innovation of AF2 with respect to its previous predecessor AlphaFold and other methods is the use of an iterative attention-based neural network architecture, which is also known as a transformer network. AF2 encodes the target protein sequence, the generated multiple sequence alignment, and structural templates as inputs to an iterative attention-based deep learning module that learns residue–residue graph edges and the sequence-residue graph edges. The residue–residue edges represent pairwise information between all residues, which can be used to predict pairwise distances or distograms. The sequence-residue edges capture the sequence evolutionary information. The system constantly updates these two representations (sequence–residue and residue–residue) by repeating this process multiple times. These representations are fed into a structure module, which also uses a transformer, that has 3D geometry built into it to produce the 3D structure and a confidence score (similar to the GDT-TS score). In the CASP14 competition, AF2 had a median GDT score of 92.4 and an approximate Root-Mean-Square Deviation (RMSD) of 1.6 Å across all targets. On the most challenging free-modeling targets, AF2′s median GDT score was 87. It has to be noted here that while AlphaFold contained a number of modules, each trained separately, whereas AlphaFold2 replaced this with a system of sub-networks coupled together into an end-to-end deep learning system trained as a single integrated structure.

In CASP14, AlphaFold2 outperformed other methods and produced remarkably accurate models that compelled organizers to declare the protein structure prediction problem for single-domain proteins to be solved and that there seems to a be a lot of enthusiasm about the possibility of solving the protein structure prediction problem to some extent in the community.

#### 3.2.4. Summary

Although existing approaches were using DL, these approaches contained a number of modules, each trained separately, whereas an end-to-end deep learning system is trained as a single integrated structure with a system of sub-networks coupled together. In that regard, end-to-end DL approaches use sequence as an input and map to protein structures end-to-end. Recent developments of end-to-end deep learning systems such as NEMO, AlQuraishi’s Recurrent Geometric Network, and AlphaFold2 for protein structure prediction represent the most remarkable advancements in the field. Co-evolutionary-related sequences, primary sequences, and PSSMs are used to predict contact map and eventually protein structure. AlQuraishi’s end-to-end model [14] predicted novel folds using primary sequences and PSSMs viz. Recurrent geometric network. AlphFold2 was very successful in the CASP14 competition by integrating the notion of attention, where the MSA matrix and primary sequence are encoded with positional embedding, which are passed to multi-head attention transformer networks to produce the protein structure. We can expect to see more end-to-end DL approaches for protein structure prediction.

## 4. Advances in Deep Learning-Based Approaches for Cryo-EM Protein Structure Determination

Owing to the significant progresses in the past decades for protein structure determination by Cryo-electron microscopy (Cryo-EM), it has evolved into one of the effective tools in structural biology. Cryo-EM is a Nobel prize-awarded technology that provides 3D maps of protein structures. The number of Cryo-EM maps deposited in EMDB as of March 2021 is ≈15 K [29], and the number of maps released is growing quite significantly (Figure 2). The fundamental computational step in this process is the interpretation of EM data to obtain protein structure information. Esquivel-Rodriguez [77] summarized advances in computational methods to model protein three-dimensional structures from a 3D EM density map that is constructed from two-dimensional maps. Based on the resolution of a Cryo-EM map, the structure information of proteins determined by Cryo-EM may not have sufficient atomic details. Generally, if the resolution of the map is less than 3 Å, atomic details can be built easily, and when the resolution is 5–10 Å (also termed as intermediate resolution), it is quite hard to get detailed structural information.

For a recent review on modeling molecular structures from density maps of different resolutions, please refer to Alnabati et al. [78] and Malhotra et al. [79]. Alnabati et al. also reviews some of the recent approaches for single particle picking and secondary structure prediction. To derive the protein structure based on its 3D Cryo-EM map, researchers either have to manually fit the atoms or resort to existing template-based or homology modeling methods. Some of the existing tools such as Rosetta, MAINMAST [80], and Phenix [81] determine only fragments of a protein complex or require extensive manual processing steps.

There are intermediate steps (viz. secondary structure prediction, backbone structure prediction, single particle picking) involved in the construction of high-resolution 3D Cryo-EM maps. Among them, single particle picking is a critical step that involves the picking of single-particle two-dimensional projections from thousands of 2D micrographs. Here, we will describe some advances in DL-based approaches for various steps in Cryo-EM based protein structure determination mainly: single-particle picking, back-bone prediction, secondary structure prediction, all-atom prediction for protein complex, and EM density map generation/refinement.

### 4.1. Deep Learning Approaches for Single Particle Picking

Cryo-EM micrographs contain 2D projections of the particles in different orientations. As a large number of single-particle images must be extracted from Cryo-EM micrographs to form a reliable 3D reconstruction of the underlying structure, particle picking is one of the critical steps and is often regarded as a bottleneck for automated structure determination from a Cryo-EM map. Recently, Deep Learning-based approaches have been emerging for single particle picking. We summarize briefly some of the existing DL-based approaches for single particle picking.

DeepPicker [82] is one of the first DL-based approaches for particle picking in Cryo-EM density maps. Tools such as DeepEM [83] and Deep Consensus [84] have been developed for single particle picking from Cryo-EM maps. Below, we describe some recent DL-based approaches for single particle picking.

#### 4.1.1. PIXER

PIXER [85] is a particle-picking method based on Deep Neural Network. One of the significant challenges in particle picking is the low signal to noise ratio. In that regard, PIXER (pixel-wise classification) uses Deep Learning-based segmentation for particle picking. Essentially, the micrographs are first converted into probability density maps using a segmentation network where probability represents the likelihood of one pixel belonging to a particle. Using a training dataset, the approach then uses the local-maximum method to identify the particles from the probability density maps. This is one of the earlier approaches that uses a segmentation network for particle selection. Comparative analysis showed that PIXER performs on par with other approaches.

#### 4.1.2. CASSPER

CASSPER [86] is another DL-based approach for particle picking. CASSPER is a semantic segmentation (SS)-based method that does pixel-level classification. CASSPER uses InceptionV4 for feature extraction and Full Resolution Residual Network architecture for semantic segmentation. Additionally, CASSPER has a Graphical User Interface (GUI) with slide bars that can be adjusted to label all particles, and this is one of the feature that distinguishes CASSPER from other existing methods. When compared with other existing tools using common datasets in the community, CASSPER was shown to achieve good performance.

#### 4.1.3. MicroGraphCleaner

MicroGraphCleaner [87] is another DL-based approach for Cryo-EM cleaning that discriminates between regions of micrographs that are suitable for particle picking and regions that are not. Essentially, it computes binary segmentation of micrographs so that the regions for particle picking can be isolated from areas containing high-contrast contaminants and other artifacts. MicroGraphCleaner is based on U-net architecture [88] and was trained on a dataset of 539 manually segmented micrographs. Benchmarking results showed that MicroGraphCleaner is a useful tool for cryo-EM cleaning. One of the salient features of MicroGraphCleaner is that it is easy to install.

#### 4.1.4. AutoCryoPicker

Although not based on DL-based approaches, it is worthwhile to mention AutoCryoPicker here. Based on three stages: image processing, particle clustering, and particle picking, AutoCryoPicker is an automated, unsupervised approach for single particle picking in Cryo-EM micrographs. One of the salient features of AutoCryoPicker is that this approach is based on an unsupervised ML algorithm: no labeled training data is required, which is sometimes hard to create in case of Cryo-EM [89].

### 4.2. Deep Learning-Based Approaches for Prediction of Backbone in Cryo-EM

For Cryo-EM images at lower resolutions such as 5–10 Å, one of the significant challenges in determining protein structures in Cryo-EM is the prediction of (location of) backbone. In this regard, recently, Si et al. developed a C-CNN (Cascaded)-based approach that comprises multiple CNNs, each predicting a specific aspect of a protein’s backbone structure viz. secondary structure elements, backbone structure, and c-alpha atoms and then combining the results to predict complete map [90]. On a benchmark set, this method performed equally well as Rosetta de novo, MAINMAST, and Phenix-based methods.

### 4.3. Deep Learning-Based Approaches for Prediction of Secondary Structures in Cryo-EM

It is still challenging to detect the secondary structure of a protein using Cryo-EM images when the spatial resolution of Cryo-EM images is at the medium level (5–10 Å). Li et al. [91] proposed a CNN-based classifier to predict the probability of labeling for every individual voxel in a 3D Cryo-EM image with respect to the a-helix, B-sheet, and background.

Recently, Kihara’s group at Purdue developed a DL-based approach called Emap2Sec [92] for detecting secondary structures of a protein in Cryo-EM maps of 5–10 Å. Emap2sec [92] first scans a Cryo-EM map with a voxel of size 11 Å. Similar to Li et al. [91], Emap2sec also uses a 3D deep-convolutional neural network that consists of a two-phase stacked network architecture. The first phase outputs probability values for an input voxel to belong to alpha-helix, Beta-sheet, or other structures, through a network with convolutional layers, a maximum-pooling layer, fully connected layers, and a SoftMax layer. The second phase network takes the probability values from the first phase as an input and outputs the final refined probabilities through a series of fully connected layers followed by a SoftMax layer. When tested on experimental Cryo-EM maps, Emap2sec showed a performance accuracy of around 64.4% on average at each amino acid level.

### 4.4. Deep Learning Approaches for All-Atom Structure of a PROTEIN COMPLEX

Here, we briefly explain the DeepTracer approach for the all-atom structure of a protein complex.

#### DeepTracer

One of the challenges in deriving the structure of a protein based on its 3D Cryo-EM map is to be able to fit the atoms to the EM map, which has to be done manually or performed using template-based or homology modeling methods. In that regard, automatically and accurately determining the molecular structure from a Cryo-EM map is an important problem.

Information about the macromolecular structure of protein complexes and related cellular and molecular mechanisms can assist the search for vaccines and drug development processes especially in the current scenario of the COVID-19 pandemic. To obtain such structural information, the authors developed DeepTracer [93], which is a fully automated deep learning-based method for fast de novo multi-chain protein complex structure determination from high-resolution Cryo-EM maps.

DeepTracer has a Deep Convolutional Neural Network, specifically U-Net, at its heart to allow for fast and accurate structure prediction and predicts four pieces of information: the location of amino acids, location of the backbone, secondary structure prediction, and amino acid types. The modified U-Net architecture in DeepTracer has four separate U-Nets, one for each structural aspect of the molecule (atoms, backbone, secondary structure elements, and amino acid types). The pre-processing step that prepares the Cryo-EM maps to be fed into the neural network primarily consists of obtaining the voxel.

Researchers applied DeepTracer on a previously published set of 476 raw experimental Cryo-EM maps and compared the results with a current state-of-the-art method. The residue coverage increased by over 30% using DeepTracer, and the RMSD value improved from 1.29 to 1.18 Å. DeepTracer determines the all-atom structure of a protein complex based solely on its Cryo-EM map and an amino acid sequence with improved accuracy and efficiency compared to previous methods.

### 4.5. Deep Learning-Based Approach for Protein Dynamics Information from Cryo-EM

Although understanding of physiological processes that proteins are involved in requires the exploration of conformational landscapes of protein complexes, structural biology until recently has been strongly driven by a static-centered view of protein architecture [94]. One of the major reasons for a lot of interest in Cryo-EM is the possibility to more broadly explore the conformational landscape of protein and protein complexes rather than a static-centered view of protein architecture.

To understand the mechanisms underlying the biological function of protein, it is essential to elucidate the 3D structure and dynamics properties of the protein. Various studies have been performed to understand/characterize dynamics of proteins. Cryo-EM is a powerful tool for the investigation of protein structures including analysis of their dynamics [95].

As protein adopts variable conformations in the specimens in Cryo-EM single-particle analysis, it can be inferred that the dynamics properties of the proteins are ‘hidden’ in the reconstructed Cryo-EM maps. Based on this hypothesis, Okuno’s group at Kyoto University developed DEFMap (Dynamics Extraction From cryo-EM Map) [96], which is a DL-based approach to extract the dynamics information of proteins from the Cryo-EM density map associated with the atomic fluctuations that are hidden in Cryo-EM density maps. The neural network used in DEFMap consists of three 3D convolutional layers with Leaky Rectified Linear Unit (ReLU) activation, max pooling, and dropout. The training data consist of a PDB model on which Molecular Dynamics (MD) simulation is performed to calculate dynamic properties (Root Mean Squared Fluctuation (RMSF) value) and the corresponding Cryo-EM data. Simultaneously, the 3D-CNN model learns the relationship between Cryo-EM map (density data) and the MD-derived RMSF. In the test phase, the trained model predicts protein dynamics values given the Cryo-EM density map. Essentially, the DEFMap using cryo-EM density data provides dynamics that correlate well with MD simulations and experimental data. DEFMap may help researchers access the dynamic properties of biological molecules.

### 4.6. Tools for Generation/Refinement of EM Map

Here, we briefly describe two tools for the generation/refinement of an EM map.

#### 4.6.1. EMRefiner

Zhang’s group recently developed EMRefiner [97], a Monte Carlo-based method for protein structure refinement and determination from Cryo-EM density maps. Although this is not a DL-based approach, this tool could be quite useful for the community. The pipeline consists of three consecutive steps of structure-to-density map superposition, rigid-body fragment adjustments, and atomic-level structure refinement. During the refinement simulations, the backbone structures are kept flexible with movements guided by a composite of physics- and knowledge-based force field, integrated with Cryo-EM density map data. The pipeline is fully automated and suitable for the protein targets with low-to-medium resolution Cryo-EM density map data.

In addition, Zhang’s group also developed DEMO-EM [98], an automatic tool to construct multi-domain structures from cryo-EM maps. As in a traditional protein structure prediction problem, the field is moving toward the determination/prediction of a multi-domain protein structure.

#### 4.6.2. SuperEm

Kihara’s group at Purdue recently developed a GAN (Generative Adversarial Network) [49]-based approach called SuperEM [99] to generate (or refine) an experimental EM Map in the resolution range of 3 to 6 Å. Owing to the success of GANs for improving other image resolution problems [100], SuperEm uses GANs to improve the resolution of experimental EM maps. The input to SuperEM is the low-resolution map, and the output is a higher-resolution map. SuperEM consists of two simultaneously trained NNs, a generator and discriminator. As in other GANs, the task here is to make the generator of SuperEM able to output high-resolution EM maps that are indistinguishable from actual high-resolution maps by the discriminator network. Simultaneously, the discriminator is trained to distinguish between the generated and real high-resolution maps. An average of 1 A resolution improvement was observed by SuperEM on a test dataset of 36 experimental EM maps. The training dataset is a low-resolution experimental EM map and a corresponding high-resolution EM map that was simulated from the associated atomic detail of the proteins. The inputs to GAN are a pair of cubes of volume (length = 25 Å) that are extracted from the EM maps. The generator network consists of a series of ResNet blocks and Discriminator consists of a series of CNN network.

#### 4.6.3. Summary

With the recent advancements in the Cryo-EM technology, Cryo-EM is becoming a leading technology for determining the protein structure. There are intermediate steps (viz. secondary structure prediction, backbone structure prediction, single particle picking) involved in the construction of a high resolution 3D Cryo-EM maps. Here, we described some DL-based advances in various steps such as single-particle picking, back-bone prediction, secondary structure prediction, EM density map refinement, and all-atom prediction for protein complex for Cryo-EM based protein structure determination.

We summarize all the methods described in the article in Table 1, highlighting the Deep Learning architecture used, the strength/uniqueness of the approach, and the availability of source code (in Github)/web server.

## 5. Future Outlook and Conclusions

We are at an exciting era in terms of protein structure prediction approaches especially due to the advancement in the field made possible by using Deep Learning. As discussed earlier, especially end-to-end Deep Learning approaches are probably going to be some of the more exciting developments in the future. Although we are likely to see advances in every aspects of protein structure pipelines over the next decade or so, we envision that the most advances will be in the following arena.

### 5.1. Better Deep Learning-Based Algorithms for MSA Generation

All protein structure prediction methods use multiple sequence alignments (MSAs) as input, and it is what seems to be driving the improvement in the accuracy of the predicted structures. Zhang’s group developed and released DeepMSA [20] as an effort to build a single pipeline for generating high-quality and deep alignments and also to find remotely homologous sequences in the case of new (difficult) sequences. Searching for homologous sequences using tools such as HHblits [30] and JackHMMER [31] against large metagenomic sequences will soon become the bottleneck in the future structure prediction methods. This is one area that we are yet to experience the vast potential of DL-based approaches. If the vast amount of sequence information in various sequence databases can be captured in a DL model, multiple sequence alignments could be generated fast, opening up a new set of possibilities to further accelerate the progress in the field.

### 5.2. Transformer Based Open-Source Approaches for Protein Structure Prediction

During the CASP14 conference in 2020, after DeepMind presented that AlphaFold2 does not use distograms or real-valued distance maps as intermediate steps for structure prediction, many researchers questioned if contact/distogram/real-valued distance maps were a wrong direction in the first place. However, since DeepMind extracted distance map predictions to investigate and explain what their models were learning, many others suggest that distogram prediction real-valued distance prediction can be extremely useful to ‘explain’ what the deep learning models have learned. During the conference, some also suggested that the distogram/distance predictions could be useful when predicted between domains. In the future, if distograms/distance maps are gradually understood to be a passage to understand what deep learning models learn, methods such as DISTEVAL [101] will also be found useful for qualitatively assessing distance maps to study the potential and limitations of structure prediction methods. A big quest in the field is to develop open-source implementations of methods that are the same or similar to AlphaFold2. With open-source and free methods, anyone can advance the research or develop commercial applications. Another quest is to learn how these advances in structure prediction can be exploited to solve other problems in the domain of protein modeling such as domain–domain docking and interaction prediction. 

### 5.3. Development of DL-Based Approaches for Multi-Domain Protein Structure Prediction

As stated by the organizers of CASP, the single-domain protein structure prediction problem is solved to some extent. In this new scenario, we are likely to see new trends in DL-based approaches for the prediction of multi-domain protein structures. In that regard, correct domain boundary assignment from sequence is a critical step toward accurate multi-domain protein structure prediction. Recently, Zhang’s group developed FUpred [102] to predict domain boundaries using contact maps and co-evolutionary precision matrices in a deep residual neural network framework. In the heart of this algorithm is the notion that domain boundaries are the locations that maximize the number of intra-domain contacts and minimize the number of inter-domain contacts. We are likely to see more methods that use DL to predict domain boundaries as well as modeling of multi-domain protein structures.

### 5.4. Explainable AI Approaches

As in other fields, the black-box nature of deep learning models has been one of the major hesitation/roadblocks for more widespread implementation/usage of DL-based approaches in Protein Structural bioinformatics. In that regard, the new trend in the field is to develop xAI (explainable AI) [103] approaches. In that regard, Cheng’s group recently developed InterPretContactMap [53] that combines deep neural network with an attention mechanism to enhance the explainability of the protein contact prediction. Not only is it very important to be able to harness the advances the power of DL algorithms to fill the existing protein sequence–structure gap, it is equally important to advance the knowledge base about the protein sequence–structure–evolution relationship. xAI may turn to be one of the approaches to arrive there.

## Figures and Tables

**Figure 1 ijms-22-05553-f001:**
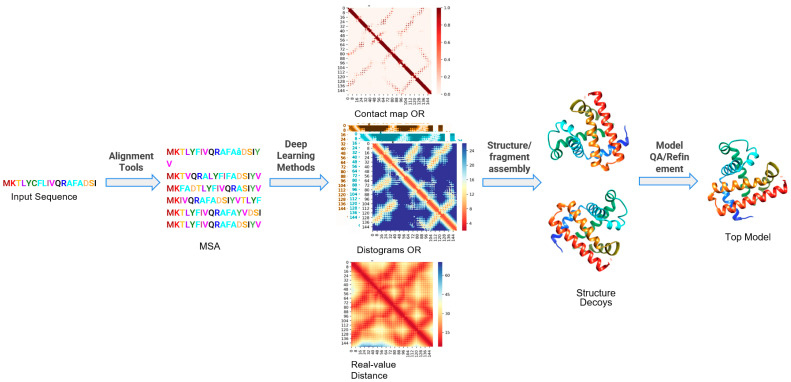
General schematic of template-free protein structure prediction pipeline. Most of the successful existing pipelines for protein structure prediction have these important steps: (i) generation of multiple sequence alignment (MSA), (ii) contact map prediction, distogram prediction or real-value distance prediction, (iii) structure/fragment assembly, and (iv) QA/refinement.

**Figure 2 ijms-22-05553-f002:**
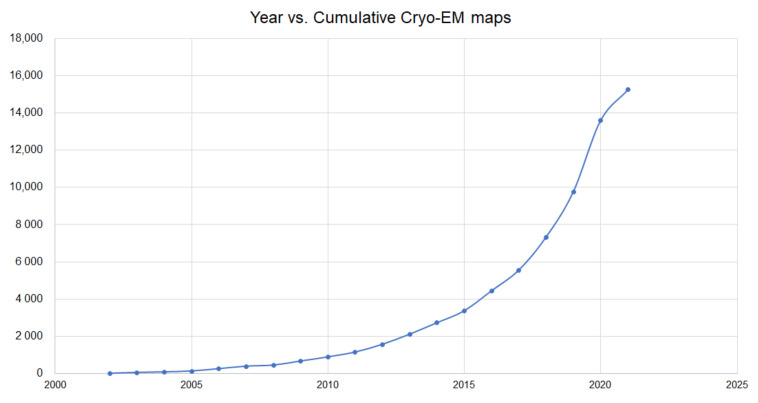
Growth of EMD density maps in EMDB from 2002 to 2020.

**Figure 3 ijms-22-05553-f003:**
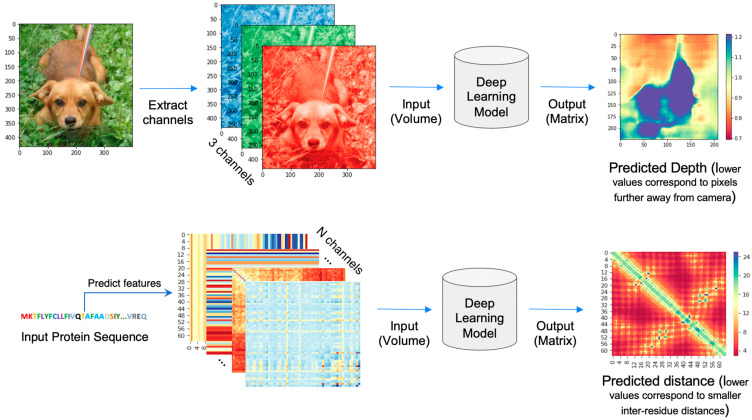
From the perspective of Deep Learning method development, the problem of protein distogram or real-valued distance prediction (bottom row) is similar to the ‘depth prediction problem’ in computer vision (top row). In all these problems, the input to the Deep Learning model is a volume (3D tensor). In case of computer vision, 2D images expand as a volume because of the RGB or HSV channels. Similarly, in the case of distance prediction, predicted 1D and 2D features are transformed and packed into 3D volume with many channels of inter-residue information.

**Table 1 ijms-22-05553-t001:** Summary of tools: category, architecture, strength/uniqueness, and availability of the tools described in this article.

Category	Tool	Architecture	Strength	Code/Web Server (Last Accessed on 10 May 2021)
End-to-end structure prediction	AlQuarishi’s end-to-end model [14]	Recurrent geometric network (RGN)	Predicted novel folds without co-evolutionary data, it achieved state-of-the-art accuracy	https://github.com/aqlaboratory/rgn
	NEMO [76]	DL	First end-to-end Deep Learning-based approach	NA
	AlphaFold 2	Transformers (attention mechanism)	Evolutionary related sequences and MSA are fetched into transformers to accurately predict protein 3D structure	NA
Real-valued distance prediction	PDNET [19]	ResNet	A fully open-source and light framework for distance, contact, and distogram prediction	https://github.com/ba-lab/pdnet/
	GAN-based method [18]	GAN+ ResNet	One of the initial efforts to predict real-valued distance maps; GANs developed to predict real-valued distance maps	https://github.com/Wenze-Codebase/DistancePrediction-Protein-GAN.git http://structpred.life.tsinghua.edu.cn/continental.html [W]
	Xu’s method [62]	ResNet	Predicts not only real-valued distance but also mean and deviation of a distance for folding	NA
	REALDIST [17]	ResNet	Highly accurate distance prediction method focusing only on real-valued distance map predictions and distance-guided 3D modeling	https://github.com/ba-lab/realdist
	DeepDist [16]	ResNet	Predicts both distograms and real-valued distances and delivers high-accuracy distance maps	https://github.com/multicom-toolbox/deepdist
Distogram (Smaller Distance range) prediction	RaptorX [11,43]	ResNet	The original RaptorX method upgraded to predict distograms	http://raptorx.uchicago.edu/AbInitioFolding/ [W], https://github.com/j3xugit/RaptorX-Contact
	ProSPr [15]	ResNet	An open-source protein distance prediction network inspired from the AlphaFold implementation	https://github.com/dellacortelab/prospr
	trRosetta [56]	ResNet	A fully Tensorflow-based open-source implementation to predict distograms; demonstrated to outperform AlphaFold	https://github.com/gjoni/trRosetta https://yanglab.nankai.edu.cn/trRosetta/ [W]
	DeepH3 [74]	ResNet	It predicts inter-residue distances and orientation from antibody heavy and light chain sequences	https://github.com/Graylab/deepH3-distances-orientations
	AttentiveDist [61]	RestNet with Attention	It uses MSAs generated with different E-values to increase the co-evolutionary information provided to the model	https://github.com/kiharalab/AttentiveDist
	DISTEVAL [101]		A tool and web server for evaluating predicted real-values distances, distograms, and contacts	https://github.com/ba-lab/disteval
Contact map prediction	QDeep [68]	ResNets	Distance-based single-model protein quality estimation method based on residue-level ensemble error classifications.	https://github.com/Bhattacharya-Lab/QDeep
	ResPRE [44]	Deep residual convolutional neural network	ResPRE is better than the methods that are built on co-evolution coupling analyses or a meta-server based neural network	https://zhanglab.ccmb.med.umich.edu/ResPRE [W], https://github.com/leeyang/ResPRE.
	MapPred [45]	Deep ResNet	Covariance features derived from MSA are used to predict contact maps, distance maps, and distance distribution	http://yanglab.nankai.edu.cn/mappred/ [W]
	DEEPCON [47]	ResNet, U-Net, and FCN	Compares various deep learning architectures for protein contact prediction	https://github.com/badriadhikari/DEEPCON/
	DeepECA [48]	CNN with ResNet	Structures predicted by DeepECA, based on contacts and SS, are more accurate than existing evolutionary coupling analysis methods	https://github.com/tomiilab/ DeepECA
	ContactGAN [50]	GAN	GAN-based denoising framework to push the limit of protein contact prediction	https://github.com/largelymfs/deepcontact
	InterPretContactMap [70]	Attention based CNN	Attention mechanisms was used to improve the interpretability of deep learning contact prediction models.	https://github.com/jianlin-cheng/InterpretContactMap
	TripletRes [54]	ResNet	TripletRes model inputs are raw co-evolutionary features, and it predicts high-accuracy contact maps	https://zhanglab.ccmb.med.umich.edu/TripletRes/ [W]
Overall protein structure prediction pipeline	AlphaFold [12]	Deep Neural Network	Accurate predictions of the distances between pairs of residues, which convey more information about the structure than contact predictions	https://github.com/deepmind/deepmind-research/tree/master/alphafold_casp13
	trRosetta [56]	ResNet	A fully Tensorflow-based open-source implementation to predict distograms; demonstrated to outperform AlphaFold	https://github.com/gjoni/trRosetta https://yanglab.nankai.edu.cn/trRosetta/ [W]
	RaptorX [11,43]	ResNet	The original RaptorX method upgraded to predict distograms	http://raptorx.uchicago.edu/AbInitioFolding/ [W], https://github.com/j3xugit/RaptorX-Contact
	MULTICOM [75]	Deep Convolutional neural network	Predicts protein structure, secondary structure, solvent accessibility, disorder region, as well as contact map	http://sysbio.rnet.missouri.edu/multicom_cluster/ [W]
	C-I-TASSER and C-QUARK [64]	Deep residual CNN	C-I-TASSER is derived from I-TASSER for high-accuracy protein structure and function predictions.	https://zhanglab.ccmb.med.umich.edu/C-I-TASSER/ [W]
Quality Assessment (QA) and refinements	QDeep [68]	ResNets	QDeep is a new distance-based single-model protein quality estimation method based on residue-level ensemble error classifications.	https://github.com/Bhattacharya-Lab/QDeep
	ResNetQA [69]	ResNet	It is a new single-model-based QA method for both local and global quality assessment.	https://github.com/AndersJing/ResNetQA
	DeepAccNet [72]	3D Convolution, 2D convolutions	DeepAccNet estimates per-residue accuracy and residue–residue distance signed error in protein models and uses these predictions to guide Rosetta protein structure refinement.	https://github.com/hiranumn/DeepAccNet
Single Particle picking or cryo-EM cleaning	PIXER [85]	Deep Neural Network	PIXER is a fully automated particle-selection method, it can acquire accurate results under low-SNR conditions within minutes.	https://github.com/ZhangJingrong/PIXER
	AutoCryoPicker [89]	Unsupervised ML algorithm	AutoCryoPicker can recognize particle-like objects from noisy Cryo-EM micrographs without the need of labeled training data, it is a useful tool for Cryo-EM protein structure determination	https://github.com/jianlin-cheng/AutoCryoPicker
	MicroGraphCleaner [87]	U-net architecture	MicrographCleaner is a tool that automatically discriminates between regions of micrographs which are suitable for particle picking, and those that are not.	https://github.com/rsanchezgarc/micrograph_cleaner_em
	CASSPER [86]	InceptionV4, Residual Network	CASSPER is the first particle picking tool implementing the Residual Network architecture for efficient pixel-wise classification.	https://github.com/airis4d/CASSPER
Structure Prediction in Cryo-EM etc.	Dong Si Method [90]	Cascade CNN	It predicts secondary structure elements, backbone structure, and Cα atoms, combining the results of each to produce a complete prediction map.	https://github.com/DrDongSi/Ca-Backbone-Prediction
	Emap2sec [92]	CNN	Emap2sec identifies the secondary structures of proteins in Electron Microscopy maps at resolutions of between 5 and 10 Å.	https://github.com/kiharalab/Emap2sec
	DeepTracer [93]	Convolutional Network Architecture	DeepTracer determines the all-atom structure of a protein complex based on a Cryo-EM map and amino acid sequence.	https://deeptracer.uw.edu/home
	DEFMap [96]	3D convolution	DEFMap directly extracts the dynamics associated with the atomic fluctuations that are hidden in Cryo-EM density maps.	https://github.com/clinfo/DEFMap
Cryo-EM	EMRefiner [97]	Monte Carlo	It is a Monte Carlo-based method for protein structure refinement and determination using a Cryo-EM density map	https://zhanglab.ccmb.med.umich.edu/EM-Refiner/
	DEMO - EM [98]	Deep Neural Network	DEMO-EM, does structure assembly of multi-domain proteins from Cryo-EM density maps.	https://zhanglab.ccmb.med.umich.edu/DEMO-EM/ [W]
	SuperEM [99]	GAN	SuperEM captures protein structure information from Cryo-EM maps more effectively than raw maps.	https://github.com/kiharalab/SuperEM
Multi Domain Protein Structures	FUpred [102]	ResNet	FUpred has better ability of domain boundary prediction than threading-based and machine learning-based methods.	https://zhanglab.ccmb.med.umich.edu/FUpred/ [W]

W: web server.

## Data Availability

Not applicable.

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
