# Peer review of "Deep Learning-Based Advances in Protein Structure Prediction"

_ijms, 2021, doi:10.3390/ijms22115553_

Round 1

Reviewer 1 Report

The authors focus on the different aspects of  Deep Learning (DL)-based approaches and on the progress that has been made so far. The provided information is very useful in the field of protein structure prediction, a hot area due quickly increasing gap between the number of known protein structures and protein sequences. At the time being, revealing the genetic information thus protein sequence has become easily accessible and effortless. In contrast determining the 3D structure is still very challenging due to difficulties in protein crystallization etc. Still understanding the 3D structure of biological molecules is a crucial element of understanding their functioning at the atomic level. The authors also highlight several points needed tto be addressed in the future to foster the development of this field.

The authors explain in a clear and concise manner why we are in desperate need for the computational prediction of protein structures. They provide highlights of the DL applications in different domains, especially in protein structure prediction.  In addition, they show the four most important steps – as a general scheme – that are needed to establish any successful pipeline for protein structure prediction. They also provide a solid background for different Deep learning based tools that are in use for the prediction of the protein structure. They discuss in detail the most recent developments in DL-based approaches for each crucial phase of the protein structure prediction pipeline. Along with the most commonly used tool, the efficiency and accuracy of each tool have been described. They also discuss the DL-based approaches for structural modeling of Cryo-EM maps.

Overall, I find this a very nice review that I suggest to be accepted after minor revision.

  1. the CASP12 abbreviation is not explained
  2. resolution of Figure 1 and 3 is very low
  3. line 86: „There have also been recent review articles highlighting 86 the Deep learning methods in protein structure prediction. No references for these review articles.” If they say this they should at least reference a few of them.
  4. It would be preferable to explain in more detail about the problem of contact map prediction
  5. They frequently refer to other review papers. It would be very nice if the authors could at least provide some concise information and knowledge as well to the manuscript. E,g, line 153 or
  6. In line 118, Prediction has to be prediction in lower case
  7. In line 129, remove an extra space

Author Response

Please find our reply to reviewer attached.

Reviewer 2 Report

This review describes the impact of deep learning methods in the field of protein structure prediction as well as in cryo-EM-based structure determination.

Below are my main concerns with this contribution

  1. first and foremost, this review lacks a compelling narrative. Some concepts are repeated over and over, such as that AlphaFold demonstrated the use of distograms, or what they are going to cover (e.g. lines 107-109 are repeated already at 116-117). Reference to the performance of the tools in the various CASP rounds is made occasionally.  Essentially all sections appear as a plain list of some relevant tools (one cannot even easily assess whether the authors are trying to be exhaustive or are just presenting a selection based on their own liking). There is no apparent logic also in the way the tools are listed - oldest to most recent, by performance or what.. Finally, there is not even a hint at informing the reader on what would be criteria for selecting a specific tool in any given situation; obviously a comparison of performance if out of scope. In short, I do not understand what is the take-home message in the authors' opinion. Personally, I would not recommend reading this to a student of mine.
  2. A few sentences in the description of tools have been copy-pasted from the original articles. E.g. DeepCon (lines 241-243 and 245-247), AttentiveDist, lines 444-445

Other relevant points are:

  • several tools presented are available as web servers. This has not been highlighted in Table 1. The authors should check that all web sites listed are indeed available and state this explicitly
  • The authors should make an effort to provide informative content for the broad biochemical readership of the journal, by explaining better how the different types of networks work and why one or the other is employed. Instead, the current version occasionally mentions information such as the number of layers for a handful of tools and one is left to wonder why is that important
  • Similarly, the authors occasionally mention average performance for some tools, such as average GDT- or TM-scores, but without a systematic comparison these numbers are completely uninformative
  • In general, the authors should make a great effort to explain basic concepts, such as what is a GDT- or TM-score at all , what they mean when they talk about L/5 or L/2 contacts.. They cannot expect that the readership of IJMS is familiar with these technical aspects

Author Response

Please find the reply to reviewer 2's comments.

Round 2

Reviewer 2 Report

I had a look at the revisions, and they seem quite reasonable. So it is publishable in my opinion.